# Spatiotemporal Heterogeneous Responses of Ecosystem Services to Landscape Patterns in Urban–Suburban Areas

Xinyan Zou [1], Chen Wang [2], Xiang Que [1,3,*], Xiaogang Ma [3], Zhe Wang [3], Quanli Fu [1,4], Yuting Lai [1,4] and Xinhan Zhuang [1,4]

1   College of Computer and Information Sciences, Fujian Agriculture and Forestry University, Fuzhou 350002, China
2   Fujian Geologic Surveying and Mapping Institute, Fuzhou 350000, China
3   Department of Computer Science, University of Idaho, Moscow, ID 83844, USA
4   Fujian Statistical Information Research Center, Fujian Agriculture and Forestry University, Fuzhou 350002, China
*   Correspondence: quexiang@fafu.edu.cn; Tel.: +86-188-5046-8868

**Abstract:** With the acceleration of urbanization, the ecosystem around cities is facing severe challenges. The drastic changes in the landscape pattern, especially in urban–suburban areas, are usually regarded as one of the main drivers. However, the spatiotemporal heterogeneous impacts of landscape patterns on the ecosystem services in this region remain unclear. To address this issue, we propose a novel framework integrating the InVEST-based ecosystem service assessment and spatiotemporal weighted regression (STWR)-based analysis of the spatiotemporal heterogeneity in urban–suburban areas, and apply it to the empirical study of Fuzhou City from 2000 to 2020. It first utilized the InVEST model to build a comprehensive ecosystem service index (CES) from five aspects (i.e., habitat quality, carbon storage, water yield, soil retention, and water purification capacity). Then, four landscape pattern indices (LPIs) (i.e., patch density (PD), area-weighted mean fractal dimension (FRAC_AM), splitting (SPLIT), and Shannon's diversity (SHDI) index) were selected to build the STWR model. We compared and analyzed the differences in the spatial coefficient surfaces and significance tests generated by the STWR model in urban, urban–suburban, and rural areas. Results show that the following: (1) The CES in Fuzhou shows an upward trend from the urban area to the urban–suburban and rural areas, with significant gradient differences. (2) Compared with other areas, the LPIs in urban–suburban areas show more fragmentation, discreteness, and diversity, indicating more socioeconomic activities. (3) Although LPIs' impacts on CES change over time (increasing from 2005 to 2010 and 2020 but decreasing in 2015), their effects are relatively low in urban–suburban areas, significantly lower than in urban areas. (4) Interestingly, the LPI coefficients near the urban–suburban boundary seem more significant. (5) This framework can effectively reveal the spatiotemporal heterogeneous relationships between various LPIs and CES, thus guiding concrete policies and measures that support decision-making for improving the ecosystem services surrounding cities through shaping landscape patterns.

**Keywords:** ecosystem services; landscape pattern; urban–suburban area; spatiotemporal weighted regression

## 1. Introduction

Ecosystem services directly or indirectly benefit humans in terms of food and raw materials, water conservation, climate regulation, and biodiversity maintenance. They can provide essential foundations for human activities and socioeconomic developments [1] . Maintaining the ecosystem's health is conducive to promoting carbon sinks and storage, as well as mitigating climate change. However, if the ecosystem is damaged, the greenhouse effect will significantly intensify [2], causing rising sea levels and even floods and other disasters [3].

Ecosystem services in cities and surrounding areas are comprehensive, complex, and affected by numerous natural and socioeconomic factors. For example, in the Kashmir Himalaya region, the invasion of alien plants (such as *Achillea millefolium*, *Aegilops tauschii*, etc.) during the urbanization process has resulted in habitat homogenization and a reduction in native species diversity, causing the degradation of habitat quality and the obstruction of ecosystem services [4]. Although we realize that the invasive species of plants and insects may affect ecosystem services in some cities, this study focuses on the impact of urbanization and socioeconomic development on ecosystem services.

Rapid socioeconomic development blocks some cities' ecosystem services, resulting in environmental issues such as pollution, habitat degradation, and diversity reduction [5,6]. The city of São Paulo, Brazil, has experienced problems such as water source, soil, and air pollution due to reduced green space and improper sewage and solid waste treatment. These environmental issues have further caused and aggravated health problems, such as respiratory diseases among residents [7].

Landscape pattern changes play an essential role in the relationships between urbanization and ecosystem services [8,9]. The variation in landscape pattern indices (LPIs), ref. [10] is closely related to the performances of ecosystem services. During urbanization, many cities neglected the protection of ecological landscapes in pursuit of rapid economic growth, resulting in severe damage to the natural landscape patterns (i.e., size, shape, and arrangement of natural ecological landscape elements) [11]. This further led to obstacles in the composition, structure, process, etc., of the ecosystem, reducing the value of its ecological services [12]. Ref. [13] found that the increase in the patch density (PD) of forests and other landscapes and the decrease in the largest patch [14] (LP) index might affect the climate regulation and water supply, resulting in the value of ecosystem services falling by approximately 60% in Baguio City, Philippines. Ref. [15] found that a more than 10% decline in the capabilities of carbon sequestration and temperature regulation of the ecosystem may be attributed to the decrease in the patch cohesion index (COHESION) and the aggregation index (AI) [16] during the rapid urbanization of Jakarta City, Indonesia.

Optimizing landscape patterns facilitates the performance-improving of ecosystem services. Many studies reported that increasing the green infrastructures [17], ecological corridors [18], and nature reserves [19] have significant positive impacts on ecosystem services. Ref. [20] found that the increase in AI, the decrease in SHDI, and the landscape shape index (LSI) [14] would be helpful to ecosystem services. Ref. [21] also found that an increase in AREA_AM (area-weighted mean patch area) and landscape division index (DIVISION) are helpful to the ecosystem [16]. Ref. [22] stated that the increase in FRAC_AM (Area-weighted mean patch fractal dimension) [14] and the decrease in SPLIT were beneficial to improving ecosystem services. Ref. [23] found that reductions in PD and edge density (ED) [14] contribute to the ecosystem.

Exploring the relationships between landscape patterns and ecosystem services during urbanization has long been a research hotspot [24]. Ref. [25] employed the multiple linear regression model to reveal the relationships in Xinjiang, China. In contrast, this model ignored the urban–rural differences [26]. Ref. [27] proposed a concentric zone gradient analysis method to explore urban–rural differences in relationships. However, this method fails to describe complex urban morphology and development trends in practical applications such as cities with multiple centers [26].

The responses of ecosystem services to landscape patterns are not just urban–rural differences but broader spatial heterogeneities. Ref. [28] utilized geographically weighted regression (GWR) [29] to explore the underlying spatial heterogeneity in Guizhou Province, China. They found that the impacts of CONTAG (Contagion Index) [16] were negative in farmland and forests, while be opposite was true in ecologically fragile areas. The SHDI (Shannon's Diversity Index) [16] also has the opposite situation in the two areas. Although the GWR method can help capture the spatial heterogeneous relationships, it fails to integrate the time dimension. The relationships in the real-world processes may not only be spatially heterogeneous but also temporally heterogeneous. Ref. [30] proposed a

novel spatiotemporal weighted regression (STWR) model that adopted a weighting strategy according to the time-varying numerical difference rate attenuation that can effectively detect the local spatiotemporal non-stationary relationships [31].

Urban–suburban areas are the most critical for landscape and ecosystem interactions. However, the spatiotemporal heterogeneity of ecosystem services' responses to landscape patterns in the areas has yet to be fully explored. Ref. [32] studied the urban–rural gradient impacts of landscape patterns on ecosystem services from 1980 to 2020 in the Beijing-Tianjin-Hebei region. They found that the most significant impact changes occurred in the urban–rural integration area. Ref. [33] also found that the ecosystem services in the urban–rural areas of Kunshan, China, have tremendous potential values, and the landscape patterns in the regions are most susceptible to changes [34]. Although some studies recognized that urban–suburban areas are critical for interaction, no clear boundary between them was defined, and their spatiotemporal patterns have not been fully explored [35,36].

To address the abovementioned concern, this study aims to develop a framework based on the InVEST and STWR models for exploring the regional spatiotemporal heterogeneity of ecosystem services' response to significant LPIs in urban–suburban areas. An empirical study in Fuzhou, China, was conducted to verify its effectiveness. Comparative analysis reveals the spatiotemporal patterns in the critical urban–suburban regions, providing insights for improving ecosystem services through optimizing landscape patterns.

## 2. Materials and Methods

### 2.1. Study Area

Fuzhou is a coastal city in the southeastern China, with coordinates ranging from 25°15′–26°39′ N and 118°08′–120°31′ E. It has a subtropical monsoon climate, abundant rainfall, multiple landscapes, and about 58.36% forest coverage [37]. Urbanization in Fuzhou has caused significant changes in landscape patterns and deterioration of the ecological environment, posing a threat to ecosystem services [38,39].

Figure 1 shows our study area. Some island areas, such as Pingtan Island, were excluded due to missing data. Drawing on the principle that building coverage decreases from downtown in urban areas to suburban and rural areas [40], we divided the study area into three categories (i.e., urban, urban–suburban, and rural areas) based on the coverage ratio of the building area. The steps for dividing the study area are as follows: First, a 900 m moving window [41] was used to calculate the coverage ratio of the building area (construction land) every five years from 2000 to 2020. Combined with the results of dividing it into three categories by the natural discontinuity method, the area with a proportion greater than or equal to 40% was considered urban. Secondly, considering that the distribution of buildings in urban–suburban areas was more widely scattered than in urban areas, the area with a ratio greater than 0 and less than 40% and its 600 m buffer zone was considered an urban–suburban area. Finally, the remaining areas were viewed as rural areas. The rationality of the classification results will be tested in subsequent experiments. Figure 1 shows the divisions of our study area in 2020.

### 2.2. Data Sources and Preprocessing

According to China's current land use classification [42], land use is divided into six categories: cultivated land, forest land, grassland, water area, construction land, and unused land. All data were unified in the coordinate system to WGS_1984_UTM_zone_50N, and ArcMap (v10.7) was used for preprocessing such as cropping, resampling, and raster calculation, and resampling to 30 m spatial resolution (Table 1).

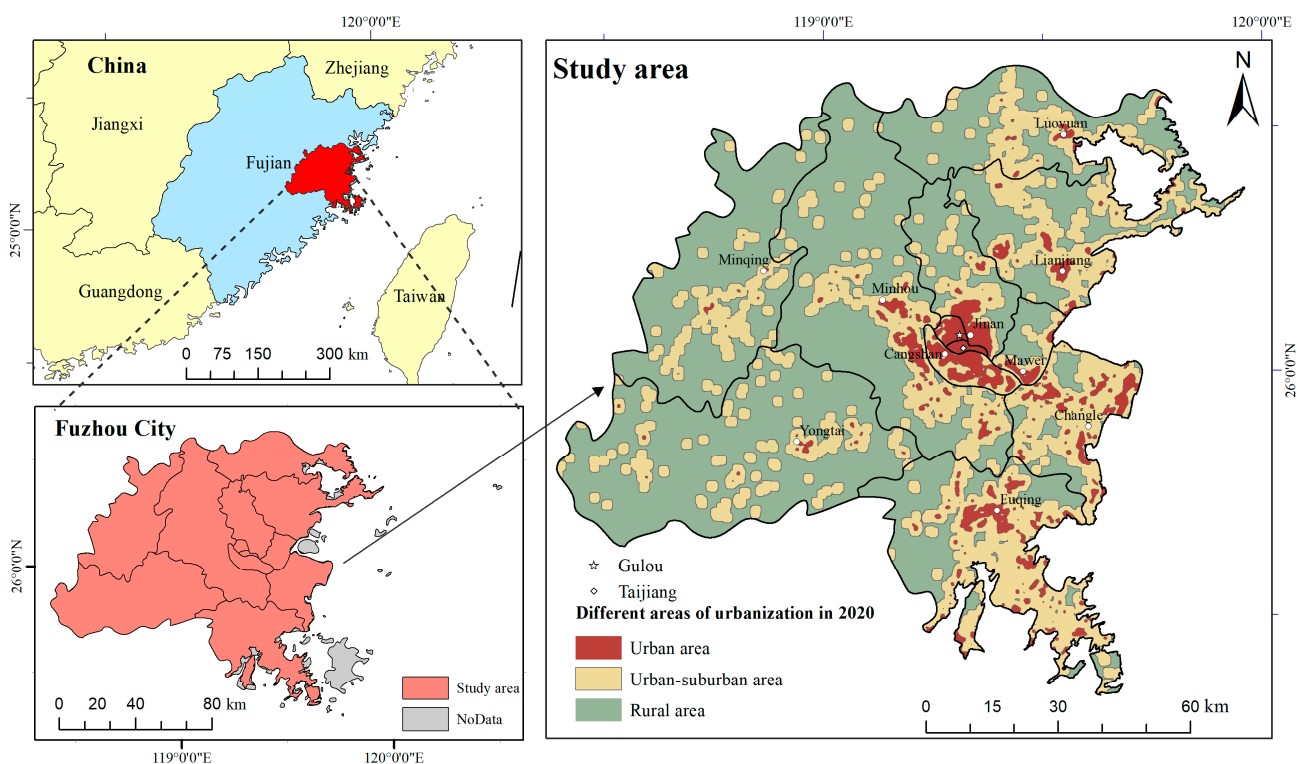

**Figure 1.** Study area (Fuzhou, China).

**Table 1.** Data sources.

| Data | Data Source | Resolution |
|---|---|---|
| Land use | Resource and Environment Science and Data Center (http://www.resdc.cn) (accessed on 13 February 2023) | 30 m |
| Precipitation | National Earth System Science Data Center (https://www.geodata.cn) (accessed on 1 March 2023) | 1 km |
| Evapotranspiration | Calculated based on the monthly potential evapotranspiration from National Earth System Science Data Center (https://www.geodata.cn) (accessed on 7 March 2023) | 1 km |
| Root restricting layer depth | depth-to-bedrock map of China in the study of [43] | 100 m |
| Plant available water content | International Soil Reference and Information Centre (ISRIC, http://data.isric.org/geonetwork/srv/eng/catalog.search#/home) (accessed on 23 March 2023) | 250 m |
| Digital elevation model | Geospatial Data Cloud (https://www.gscloud.cn/) (accessed on 21 March 2023) | 30 m |
| Normalized difference vegetation index (NDVI) | National Aeronautics and Space Administration (NASA, https://search.earthdata.nasa.gov/search) (accessed on 14 April 2023) | 1 km |
| Soil type and soil texture | Harmonized the World Soil Database (HWSD, https://www.fao.org/soils-portal/data-hub/soil-maps-and-databases/harmonized-world-soil-database-v12/en/) (accessed on 8 March 2023) | 1 km |

*2.3. Research Methods*

2.3.1. Ecosystem Services Assessment

The assessment of the comprehensive index for ecosystem services (CES) is based on five ecosystem services evaluated using the InVEST model, including habitat quality, carbon storage, water yield, soil retention, and water purification capacity. It is used

to reflect the total supply of multiple ecosystem services in the city. The formula was defined as [44]:

$$CES_j = \sum_{i=1}^{n} w_i \cdot s_{ij} \tag{1}$$

where $CES_j$ is the comprehensive index for ecosystem services in the j-th year; $I_i$ is the weight of the i-th ecosystem service; $s_{ij}$ is the normalized value of the i-th ecosystem service in the j-th year; n is the number of types of ecosystem services.

Using the InVEST model, the habitat quality was assessed by correlating land use and land cover (LULC) with stress factors, subsequently used to evaluate biodiversity support service functions. The total carbon storage was assessed considering the four major parts: aboveground carbon storage, underground carbon storage, soil carbon storage, and dead organic matter carbon storage. The water yield was calculated based on the Budyko hydrothermal coupled equilibrium assumption [45–47], and it can be regarded as the difference between precipitation and actual evapotranspiration. The soil retention was calculated according to the revised universal soil loss equation [48] based on the difference between potential and actual soil erosion while considering sediment retention. The water purification capacity was evaluated by the retention and output of nitrogen and phosphorus nutrients (only non-point source pollution is considered) [49].

### 2.3.2. LPIs

LPIs are widely used to quantify the characteristics of landscape patterns [50]. Table 2 lists 16 normally used LPIs at the landscape level and their basic descriptions [14,16,51].

**Table 2.** LPIs and their corresponding description.

| Structural Feature | Index | Expression | Description |
|---|---|---|---|
| Area/density/edge | Number of patches (NP) | $NP = N$ | N = total number of patches in the landscape. A = total landscape area (m²). E = total length (m) of edge in landscape. $a_{ij}$ = area (m) of patch ij. $E^*$ = total length (m) of edge in landscape. |
| | Patch density (PD) | $PD = \frac{N}{A}(10,000)(100)$ | |
| | Edge density (ED) | $ED = \frac{E}{A}(10,000)$ | |
| | Mean patch area (AREA_MN) | $AREA\_MN = \frac{A}{N}$ | |
| | Largest patch index (LP) | $LP = \frac{max(a_{ij})}{A}(100)$ | |
| | Landscape shape index (LSI) | $LSI = \frac{25E^*}{\sqrt{A}}$ | |
| Shape | Area-weighted mean patch fractal dimension (FRAC_AM) | $FRAC\_AM = \sum_{i=1}^{m}\sum_{j=1}^{n}\left[\left(\frac{2\ln(0.25P_{ij})}{\ln a_{ij}}\right)\left(\frac{a_{ij}}{A}\right)\right]$ | $P_{ij}$ = perimeter (m) of patch ij. $c_{ijr}$ = contiguity value for pixel r in patch ij. v = sum of the values in a 3-by-3 cell template. $a_{ij}^*$ = area of patch ij in terms of number of cells. |
| | Area-weighted mean patch shape index (SHAPE_AM) | $SHAPE\_AM = \sum_{i=1}^{m}\sum_{j=1}^{n}\left[\left(\frac{0.25P_{ij}}{\sqrt{a_{ij}}}\right)\left(\frac{a_{ij}}{A}\right)\right]$ | |
| | Mean contiguity index (CONTIG_MN) | $CONTIG\_MN = \frac{\sum_{i=1}^{N}\left[\frac{\left[\frac{\sum_{r=1}^{z}c_{ijr}}{a_{ij}^*}\right]-1}{v-1}\right]}{N}$ | |
| Connectivity | Patch cohesion index (COHESION) | $COHESION = \left[1 - \frac{\sum_{i=1}^{m}\sum_{j=1}^{n}P_{ij}^*}{\sum_{i=1}^{m}\sum_{j=1}^{n}P_{ij}^*\sqrt{a_{ij}^*}}\right] \cdot \left[1 - \frac{1}{\sqrt{Z}}\right]^{-1} \cdot (100)$ | $P_{ij}^*$ = perimeter of patch ij in terms of number of cell surfaces. Z = total number of cells in the landscape. |
| Contagion/interspersion | Aggregation index (AI) | $AI = \left[\sum_{i=1}^{m}\left(\frac{g_{ij}}{max \to g_{ij}}\right)P_{ij}\right](100)$ | $g_{ij}$ = number of like adjacencies between pixels of patch type i based on the single-count method. $max - g_{ij}$ = maximum number of like adjacencies between pixels of patch type i based on the single-count method. $P_i$ = proportion of the landscape occupied by patch type i. $g_{ik}$ = number of adjacencies between pixels of patch types i and k based on the double-count method. m = number of patch types present in the landscape. |
| | Landscape division index (DIVISION) | $DIVISION = \left[1 - \sum_{i=1}^{m}\sum_{j=1}^{n}\left(\frac{a_{ij}}{A}\right)^2\right]$ | |
| | Contagion index (CONTAG) | $CONTAG = \left[1 + \frac{\sum_{i=1}^{m}\sum_{k=1}^{m}\left[P_i \cdot \frac{g_{ik}}{\sum_{k=1}^{m}g_{ik}}\right]\cdot\left[\ln\left(P_i \cdot \frac{g_{ik}}{\sum_{k=1}^{m}g_{ik}}\right)\right]}{2\ln m}\right](100)$ | |
| | Splitting index (SPLIT) | $SPLIT = \frac{A^2}{\sum_{i=1}^{m}\sum_{j=1}^{n}a_{ij}^2}$ | |
| Diversity | Shannon's evenness index (SHEI) | $SHEI = \frac{-\sum_{i=1}^{m}(P_i \cdot \ln P_i)}{\ln m}$ | |
| | Shannon's diversity index (SHDI) | $SHDI = -\sum_{i=1}^{m}(P_i \cdot \ln P_i)$ | |

### 2.3.3. STWR

STWR is a temporally extended model based on the GWR. It is characterized by considering the numerical time-varying attenuation weighting strategy and can accurately describe the impact of nearby points in space and time on the regression point [52]. The basic formula of the STWR model can be defined as [53]:

$$y_i^t = \beta_0^t(u_i, v_i) + \sum_k \beta_k^t(u_i, v_i)x_{ik}^t + \varepsilon_i^t \tag{2}$$

$$\hat{\beta}_k^t(u_i, v_i) = \left(X^T W^t(u_i, v_i)X\right)^{-1} X^T W^t(u_i, v_i)y^t \tag{3}$$

where $y_i^t$ is the response variable of the regression point whose spatial position is $(u_i, v_i)$, $x_{ik}^t$ is the dependent variable, and $\varepsilon_i^t$ is the error term. $W^t$ is a diagonally weighted matrix calibrated by a specified kernel function of a given bandwidth. Each element in $W^t$ reflects the influence of another observation point on the regression point.

### 2.3.4. Research Framework

Figure 2 shows our designed framework for exploring the spatiotemporal heterogeneities between LPIs and ecosystem services in urban–suburban areas. This framework consists of two parts: (1) Assessment of the ecosystem services based on the InVEST model. (2) Comparative analysis of the spatiotemporal heterogeneity in different areas based on the STWR model.

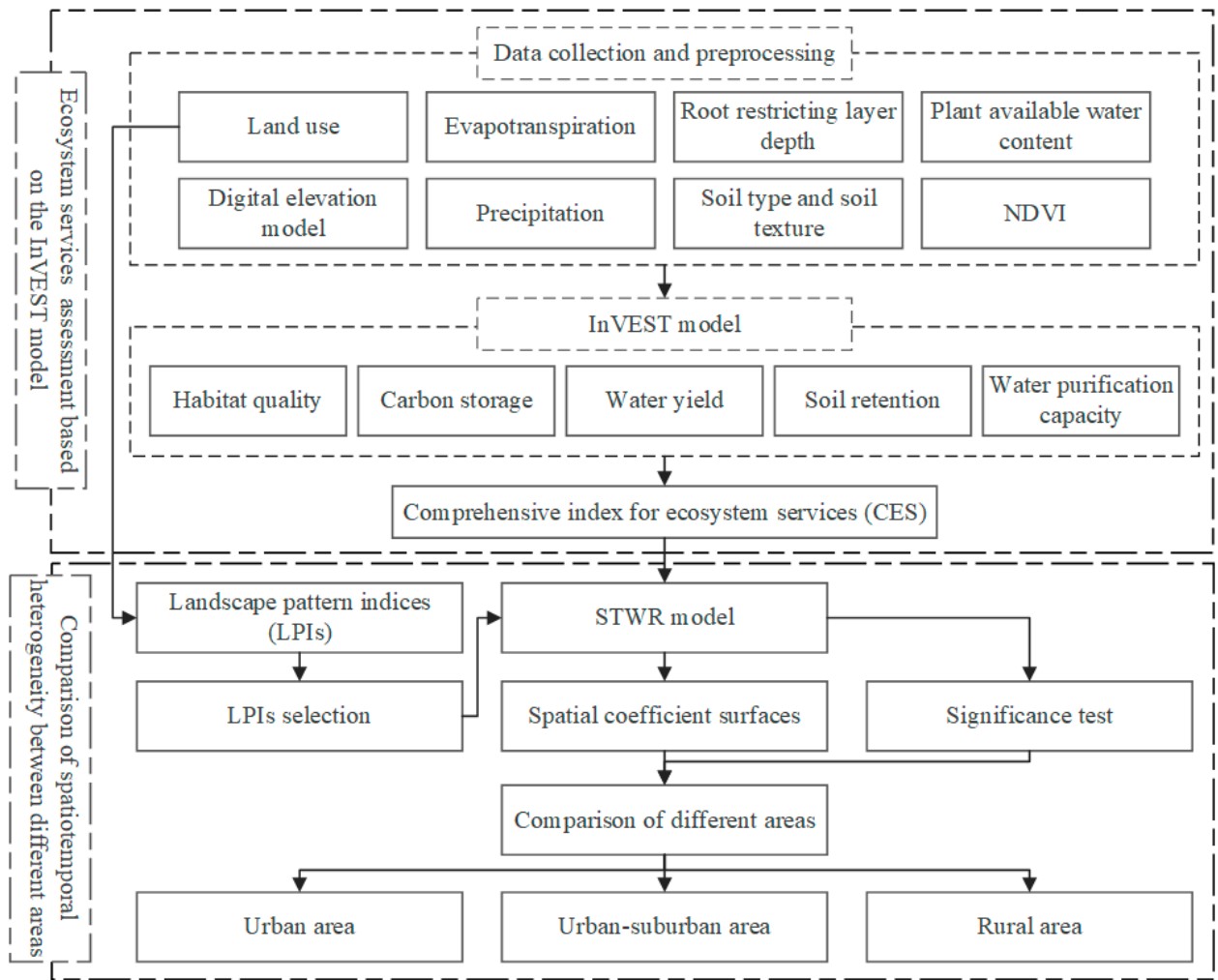

**Figure 2.** Research framework.

### 2.3.5. Parameters Settings

In measuring ecosystem services, habitat quality was evaluated with the parameter settings of relevant threat factors, habitat types, sensitivity table, threats table, and model parameters referring to [54]. The carbon storage was evaluated using the carbon density data referring to [55]. Water yield was assessed by referring to the main calculation formulas, process, and biophysical table in the Invest model operation manual [56]. Among them, the seasonality factor was repeatedly verified based on the average water production coefficient of the total water resources in the "Fuzhou Water Resources Bulletin", and the final value was 2.35. Evaluating soil retention was referred to [57]. The calculation of rainfall erosivity and soil erodibility were referred to [58] and [59], respectively. The calculation of the cover–management factor is based on the NDVI [60]. The support practice factors were referred to [61]. The water purification capacity was evaluated using the parameter and biophysical table from [62].

### 3. Results

All the LPIs mentioned in Table 2 were calculated using the moving window method in Fragstats (v4.2) software [14]. The LPIs with a correlation coefficient with CES of less than 0.2 were excluded. Then, a recursive multicollinearity diagnosis was performed, gradually excluding the LPI variables with a variance inflation factor (VIF) [63] greater than 6 in each loop. Finally, four stable LPIs (i.e., PD, FRAC_AM, SPLIT, and SHDI) with all their VIF under six were selected for further analysis.

### 3.1. CES Assessment

To assess the comprehensive ecosystem service (CES) in our study areas, we first utilized the InVEST model to evaluate five specific ecosystem services: habitat quality, carbon storage, water yield, soil retention, and water purification capacity. Figure 3 compares their trends in urban, urban–suburban, and rural areas. For habitat quality, carbon storage, and soil retention services, an increasing trend can be identified from urban to urban–suburban to rural areas. This may indicate that the levels of these three ecosystem services are higher in areas with higher vegetation coverage, while the opposite is true in areas with lower vegetation coverage. The evaluated water yield service was higher in urban than urban–suburban and rural areas, which might be related to the higher proportion of impervious surfaces and lower evapotranspiration in urban areas [44]. The average evaluated nutrient export (nitrogen and phosphorus) [64] illustrated a decreasing trend from urban to urban–suburban and rural areas, indicating an opposite order in the water purification capacity. This may be attributed to urban and urban–suburban areas being prone to environmental pollution and eutrophication issues, resulting in relatively weak water purification capabilities. In contrast, more significant areas of forestland in the suburbs can reduce nutrient output [65,66].

From 2000 to 2020, the spatial distribution of CES showed heterogeneity, with significant changes along the urban–rural gradient (Figure 4). From 2000 to 2020, the spatial distribution of CES showed heterogeneous characteristics: Low-CES areas were primarily concentrated in urban areas, while High-CES areas were mainly concentrated in rural areas. Some spatial spillover phenomena could also be detected in the low-CES areas. In addition, the average CES illustrated an increasing trend from urban to suburban and rural areas. This may be because buildings with low ecosystem service benefits dominate land use in urban areas. However, there were fewer buildings in suburban and rural areas [37], and the landscape is mainly composed of cultivated land, grassland, and woodland with high ecosystem service benefits [51]. This finding is consistent with the results of [26,33].

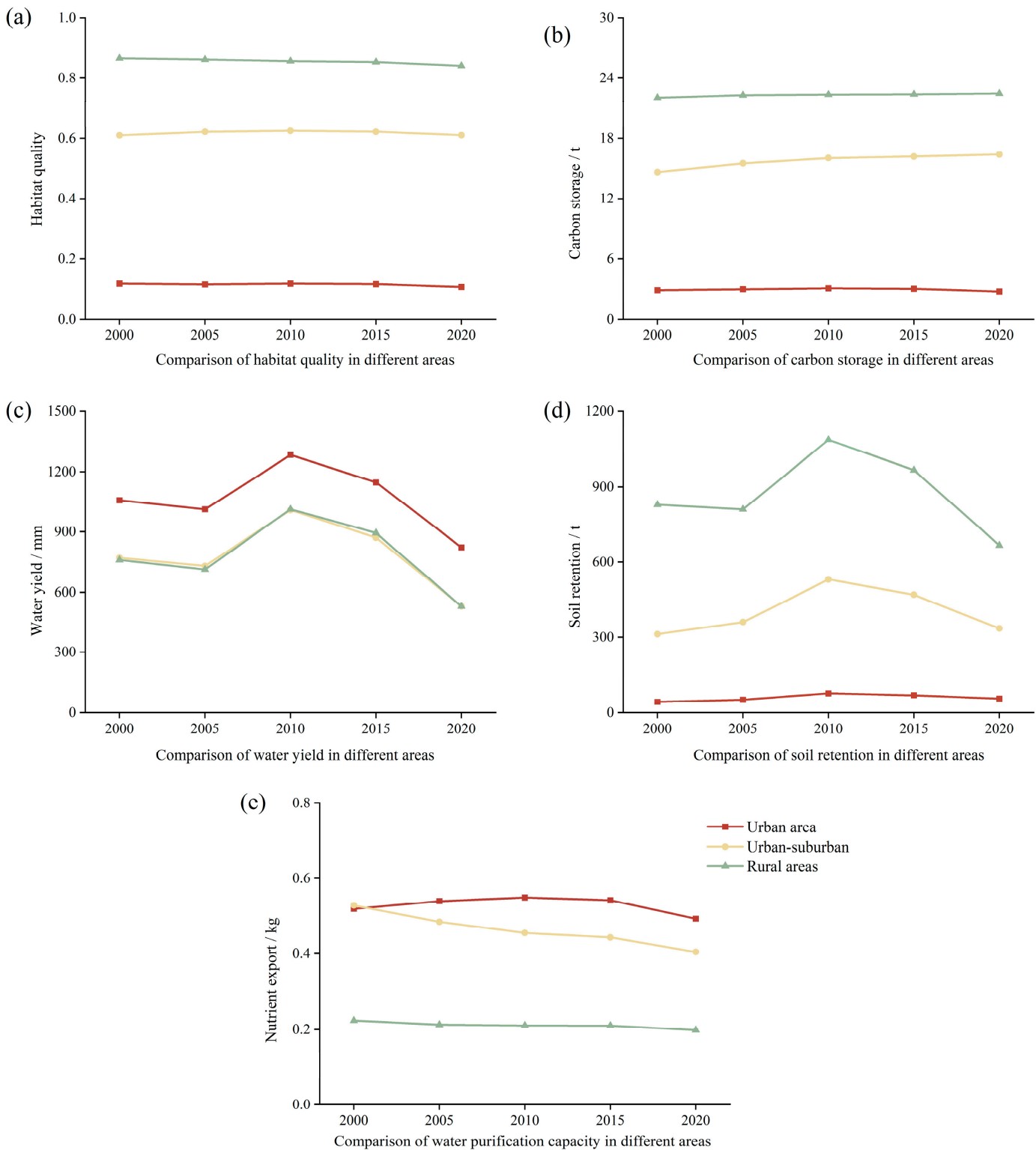

**Figure 3.** Comparison of average ecosystem service capabilities of (**a**) habitat quality, (**b**) carbon storage, (**c**) water yield, (**d**) soil retention, and (**e**) water purification capacity in different areas from 2000 to 2020.

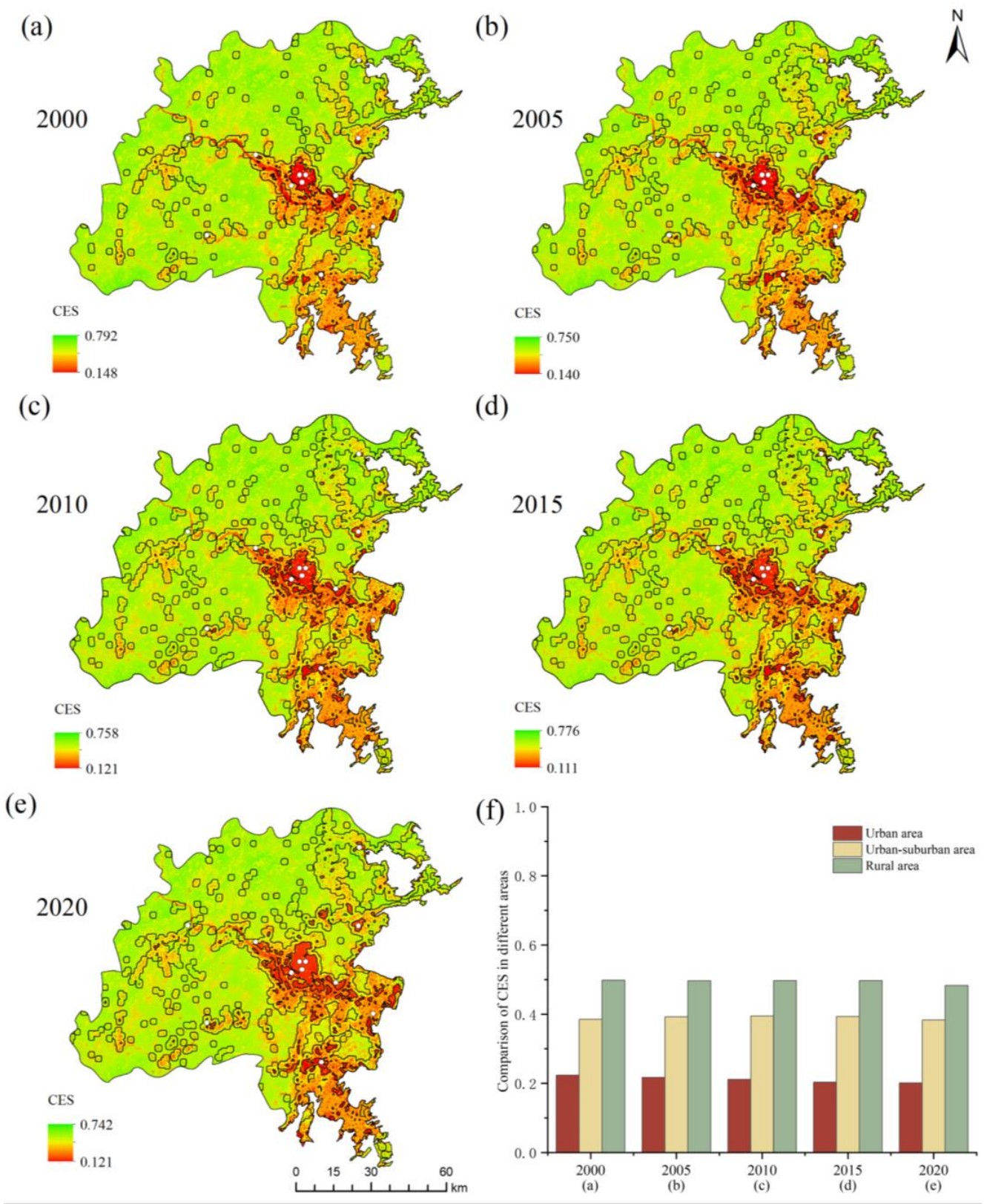

**Figure 4.** Spatial distribution of CES in (**a**) 2000, (**b**) 2005, (**c**) 2010, (**d**) 2015, and (**e**) 2020, and (**f**) mean CES in different areas.

### 3.2. LPIs

As shown in Figure 5, the LPIs in urban–suburban areas are significantly higher than in other regions, indicating higher vulnerability, fragmentation, discretization, diversity, and, therefore, more socioeconomic activities. The rural area's landscape pattern is characterized by higher aggregation, more regular patch shapes, and lower diversity, which indicates fewer socioeconomic activities. This is similar to the findings in [26], which stated that PD, SHDI, and SPLIT were most prominent in urban fringe areas. The impacts of FRAC_AM were highest in urban–suburban areas, indicating the most complex patch shape throughout the study area. This finding is also consistent with [67].

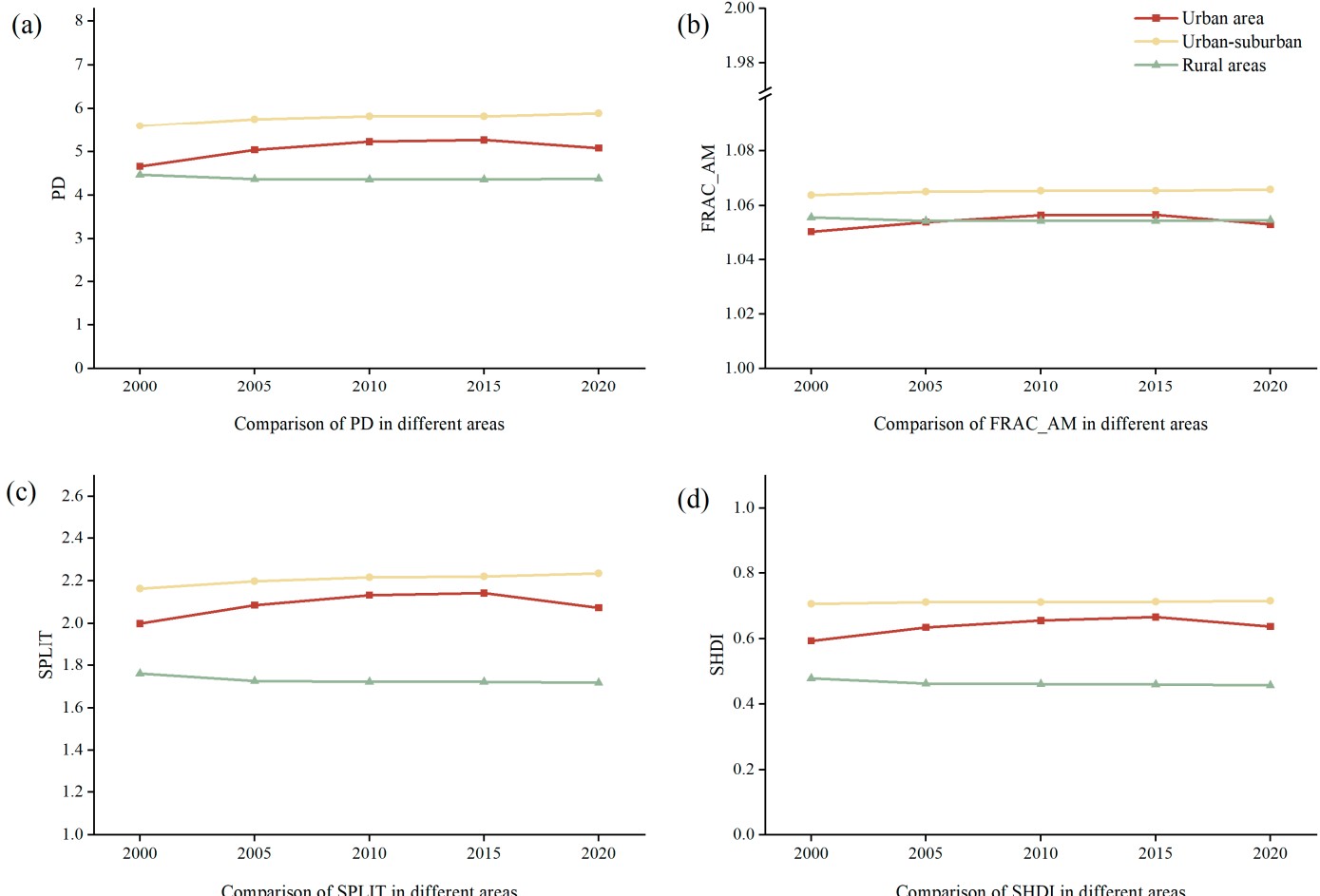

**Figure 5.** Comparison of average (**a**) PD, (**b**) FRAC_AM, (**c**) SPLIT, and (**d**) SHDI in different areas from 2000 to 2020.

### 3.3. Spatiotemporal Heterogeneous Response of CES to LPIs

To further explore the impacts of LPIs on CES, 1000 random points were sampled to build the OLS, GWR, and STWR models. Table 3 shows the comparison results: The STWR model outperformed the OLS and GWR model with higher $R^2$ and lower RSS and AICc from 2000 to 2020.

**Table 3.** Comparison of OLS, GWR, and STWR model results.

| Years | Model | $R^2$ | RSS | AICc | Sigma | Optimal Spatial (Temporal) Bandwidth |
|---|---|---|---|---|---|---|
| 2000 | OLS | 0.182 | 8.434 | −1925.497 | 0.092 | — |
| | GWR | 0.195 | 8.298 | −1928.817 | 0.092 | 939 |
| | STWR | 0.684 | 3.259 | −2363.117 | 0.064 | 61 (1) |
| 2005 | OLS | 0.179 | 8.819 | −1880.834 | 0.094 | — |
| | GWR | 0.192 | 8.688 | −1882.160 | 0.094 | 921 |
| | STWR | 0.977 | 0.249 | −4226.276 | 0.013 | 8 (2) |
| 2010 | OLS | 0.172 | 9.856 | −1769.676 | 0.099 | — |
| | GWR | 0.182 | 9.742 | −1769.936 | 0.099 | 997 |
| | STWR | 0.976 | 0.292 | −4524.266 | 0.010 | 7 (3) |
| 2015 | OLS | 0.176 | 10.590 | −1697.908 | 0.103 | — |
| | GWR | 0.188 | 10.441 | −1698.367 | 0.103 | 921 |
| | STWR | 0.928 | 0.930 | −3588.073 | 0.018 | 12 (3) |
| 2020 | OLS | 0.164 | 10.207 | −1734.759 | 0.101 | — |
| | GWR | 0.177 | 10.048 | −1735.070 | 0.101 | 888 |
| | STWR | 0.949 | 0.617 | −4010.743 | 0.013 | 8 (4) |

Figure 6 shows the impacts of four LPIs in different areas. The impacts of LPIs in urban–suburban areas were relatively low, especially significantly lower than those in urban areas. The impact of LPIs in urban–suburban areas increased from 2005 to 2010 and 2020, while the effect was the opposite in 2015. Among all LPIs, the SPLIT had a relatively more prominent negative impact on CES in urban–suburban areas. Specifically, the impact of each LPI on CES was as follows:

(1) The impacts of PD in the three areas were similar before 2015, while the positive effects have mainly manifested in urban areas since then.

(2) The impacts of FRAC_AM illustrate a trend from positive to negative along the urban–rural gradient. In urban areas, FRAC_AM is relatively strongly correlated with CES. The positive effects weakened in urban–suburban areas and showed negative impacts in rural areas.

(3) The impacts of SPLIT were like FRAC_AM and primarily positively correlated to CES in urban areas, while slightly opposite in urban–suburban and rural areas. In urban areas, the increase of SPLIT is beneficial to improving its CES to a certain extent. In contrast, in urban–suburban regions and rural areas, the decrease of SPLIT has a specific promoting effect on improving CES.

(4) SHDI's impacts differed from those of other LPIs, mainly showing negative impacts on CES. The negative effects were most significant in urban areas, while they weakened in urban–suburban and rural areas. In 2020, the mean coefficient climbed above 0 in rural areas, indicating that high-quality landscapes such as woodlands and grasslands are concentrated, and higher SHDI promotes the increase in CES. This finding was also consistent with the results of [28].

Figure 7 shows the spatial distribution of the significant coefficients of LPIs in 2005 and 2015. Interestingly, the substantial impacts of LPIs on CES are primarily concentrated in urban–suburban and rural areas, especially at many significant points close to the urban–suburban boundaries. This also indicated that as urbanization expands, the change in urban–suburban areas plays an essential role in the impact of LPIs on CES. Specifically, in 2015, the number of significant PD coefficient points increased, with positive coefficients mainly concentrated close to the urban–suburban boundaries or rural areas. The negative significant coefficients were primarily focused in the east regions close to the urban area, indicating that the urbanization expansion may cause an increase in PD, thus resulting in negatively affected CES. The negative significant coefficients of FRAC_AM were mainly concentrated at the urban–suburban boundaries of the northwest and southwest areas. The positive significant coefficients were focused on the central regions. Significant SPLIT

coefficients increased in 2015, with negative coefficients primarily distributed in the western rural areas with large landscape patches and positive coefficients concentrated in the eastern regions closer to the urban area. Significant negative SHDI coefficients were primarily focused in the central area near the urban areas. SHDI-positive coefficients were mainly distributed in the northern and southern areas near the urban–suburban boundaries.

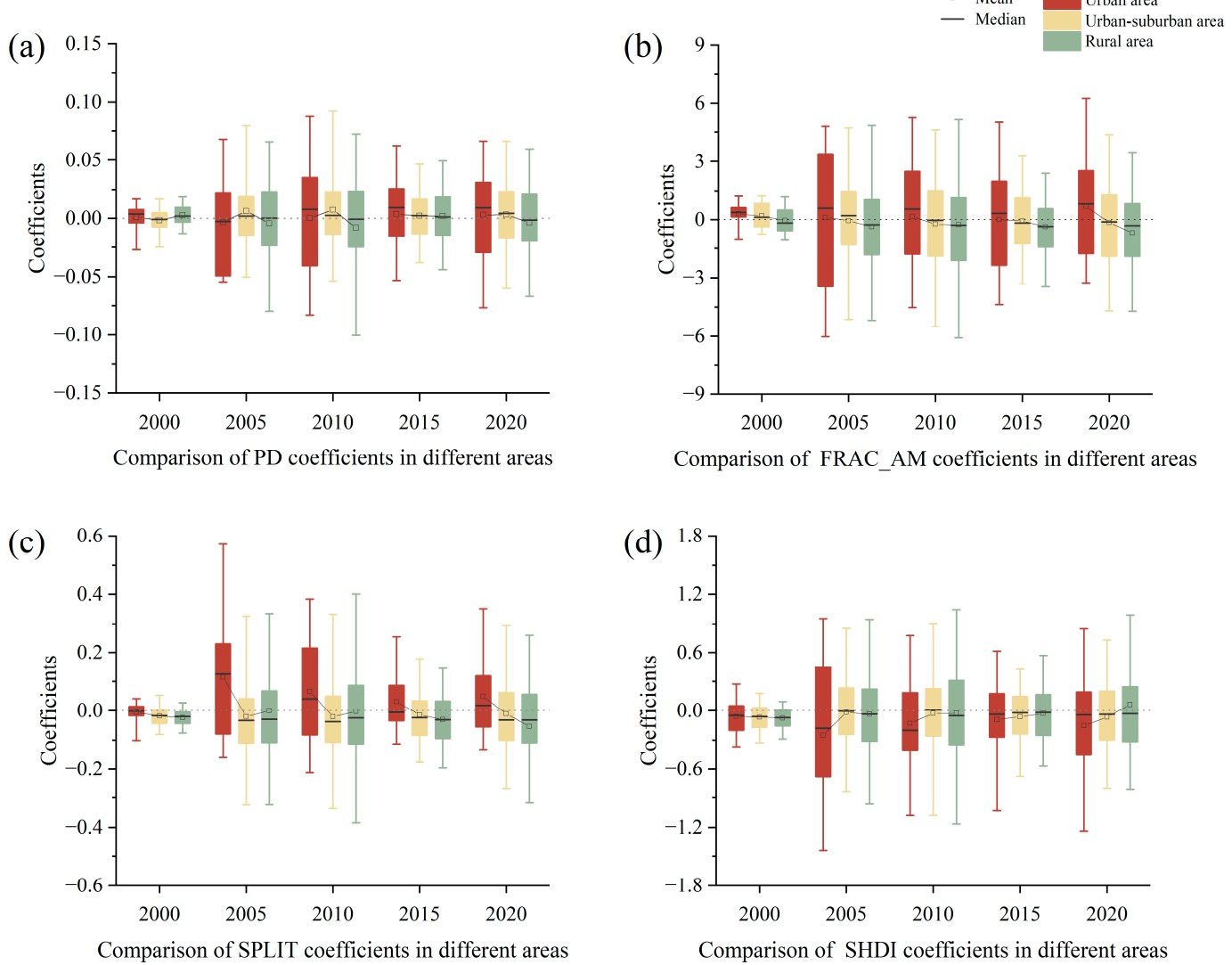

**Figure 6.** Comparison of (**a**) PD, (**b**) FRAC_AM, (**c**) SPLIT, and (**d**) SHDI coefficients generated by the STWR model in different areas from 2000 to 2020.

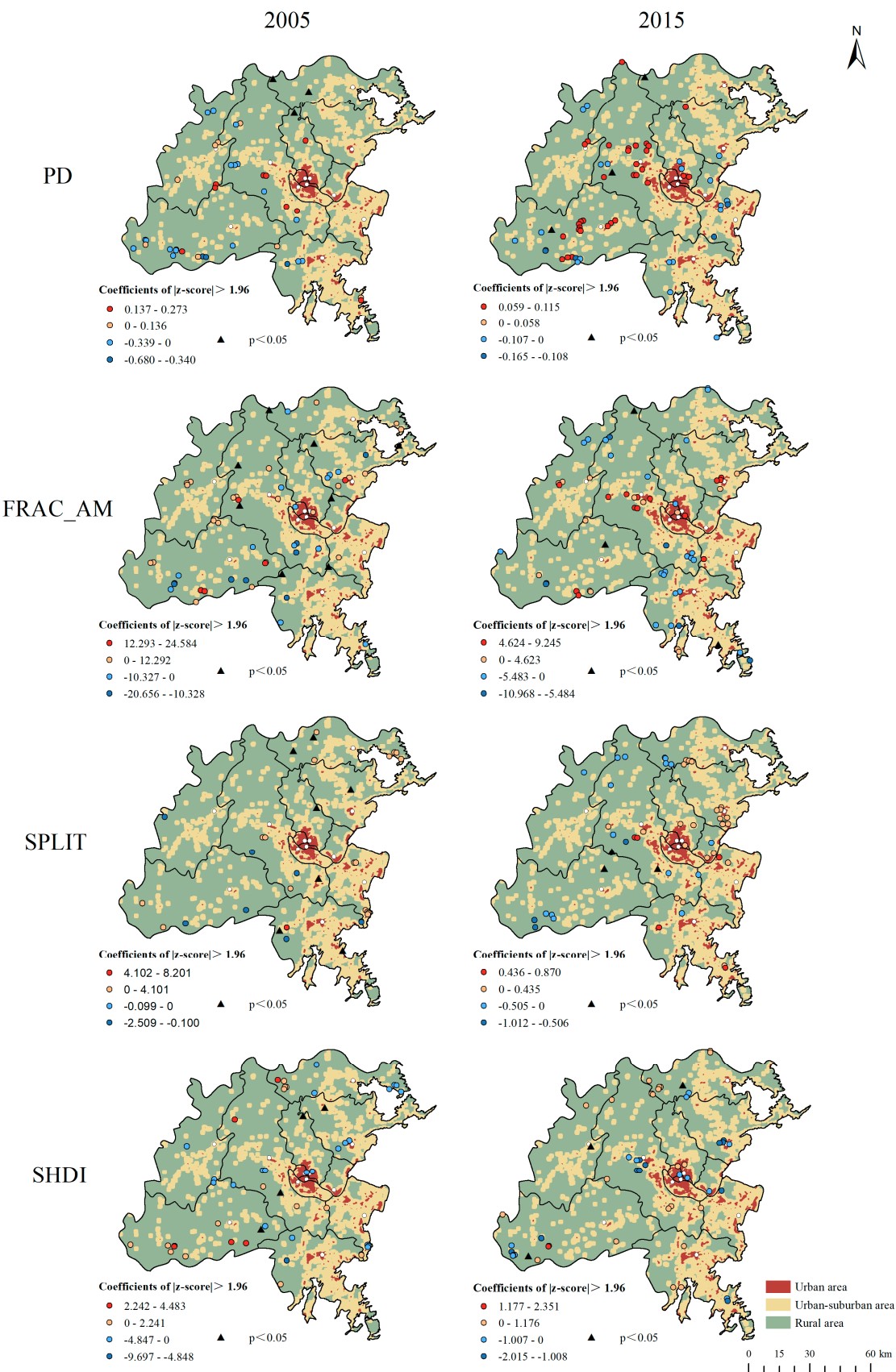

**Figure 7.** Significant coefficients of LPIs in 2005 and 2015.

## 4. Discussion

Empirical studies demonstrated that our framework is effective in real-world applications. In addition to exploring the characteristics of changes along the urban–rural gradient, our framework also provides more refined spatiotemporal heterogeneity, helping to explore local spatiotemporal patterns between landscape patterns and ecosystem services in critical urban–suburban areas:

(1) In urban-suburban areas, the impact of LPIs on CES increased from 2005 to 2010 and 2020, but the effect weakened in 2015. Specifically, PD turned from slightly negative to weak positive in 2005. The other three indices (i.e., FRAC_AM, SPLIT, and SHDI) illustrated weak negative and somewhat weakened since 2005. These indices are generally closer to 0 than in urban and rural areas, implying that the urban–suburban regions may have a unique "buffering" effect, causing the coefficients of urban and suburban areas to "cancel" each other. In addition, the increased impact may be related to the gradual increase in urbanization after 2004. The period from 2004 to 2010 was the main stage when Fuzhou's built-up areas expanded to the west. From 2015 to 2020, Fuzhou accelerated the construction of the Binhai New Development Zone and other projects, a period of rapid urbanization growth [24,39].

(2) Although landscape pattern shows significant vulnerability, fragmentation, discretization, and diversity, indicating more socioeconomic activities in urban–suburban areas (Figure 5), the coefficients of LPIs in this area were lower than those of urban areas (Figure 6). This may be attributed to urban–suburban areas having higher-quality landscapes [68], such as cultivated land, grassland, and woodland, than urban areas. These landscapes had stronger resistance to the degradation of habitat quality caused by urban expansion [69]; thus, to a certain extent, the impacts of LPIs on CES were alleviated.

(3) Among all LPIs, the SPLIT has a more substantial negative impact on CES. SPLIT (Table 2) calculates the proportion of the square of the total landscape area to the sum of the squares of all patch areas in the landscape. It reflects the degree of fragmentation of the landscape. This indicates that urban expansion may cause the fragmentation of large high-quality landscape patches, such as forestland, cultivated land, and grassland, to be encroached upon and divided into smaller patches, adversely affecting the composition, structure, and ecosystems, thereby leading to the decline of CES. Ref. [39] reported that the construction land occupied a large amount of green space during urbanization in Fuzhou from 2000 to 2020, which led to higher discreteness of high-quality landscapes, thus weakening the stability of their ecosystems. Accordingly, ecological network construction [70] can be optimized to reduce the discreteness of high-quality landscapes, thereby improving ecosystem services.

(4) A large number of significant points were close to the boundaries of urban–suburban areas (Figure 7). Regarding this finding, we believe the following aspects are worthy of consideration: First, urban–suburban boundaries change dynamically, and the situation near their boundaries can be complex. Urban–suburban areas alleviate the supply and demand problems of ecosystem services in urban areas [71], and these areas also reduce the pressure of urban expansion on the ecological environment of suburbs [72]). The near urban–suburban boundary areas were the first and most directly affected by urban expansion's speed, direction, and form. Theoretically, it should indeed be more significantly affected than other areas. These significant points surrounding the boundaries, to a certain extent, reflected the rationality of our method. However, these points do not reflect the specific speed, direction, and form of urban expansion. Thus, we cannot directly tell how the specific urban expansion processes affected these areas' ecosystem services (CES). Further research could combine specific urban expansion policies and explore the changes in significant coefficients to identify more refined patterns. Second, some of these significant coefficient points close to the urban–suburban boundaries were positive, and some were negative. This seems to imply that landscape changes caused by urbanization should have two opposite

effects on ecosystem services. If landscape shaping is appropriately handled during the expansion of urbanization, it is possible to transform ecosystem services for the better. Third, the urban–suburban boundaries may play a particular "resistance" role. We found that the overall absolute values of the LPIs' coefficients in the suburban area were smaller than those in the urban area. However, whether or how these significant points close to the boundaries were related to the suburban landscape's "resistance" to the "invasion" of urbanization remains unclear. It also reminds us that the changes in urban–suburban areas were critical for the CES's response to LPIs during urbanization. Therefore, the urban–suburban area, especially near its boundary, still needs further research and attention.

(5)  The definition of the boundaries of urban–suburban areas still lacks broad, unified standards [73]. Although our divisions of urban and suburban areas obtained some reasonable results, the in-depth exploration of the spatiotemporal heterogeneous relationships between LPIs and CES in areas close to the boundaries is still in its infancy and limited. Future research could adopt a more scientific division method.

In addition, this work can provide visual decision-support references for relevant governments and departments, including, but not limited to, the Ecological Environment Bureau, Forestry Bureau, Natural Resources and Planning Bureau, Urban and Rural Development Bureau, and Garden Department. Some concrete measures derived from our empirical analysis are listed below: (I) In urban–suburban areas, especially areas close to the boundaries, local governments should make scientific plans for the building areas, avoiding blind expansion that may cause damage to large tracts of high-quality landscapes, such as cultivated land, woodland, and grassland, thus reducing the increase in FRAC_AM, SPLIT, and SHDI. Plantations and grassland can be added near the urban–suburban boundary areas as buffer zones to improve regional landscape PD. Governments and relevant departments should pay attention to constructing and optimizing environmental networks between urban and rural areas. (II) In urban areas, we can add "green" to areas with dense construction land to promote the increase of PD, FRAC_AM, and SPLIT, thus improving CES. Some areas with sparse construction land can be merged to reduce fragmentation and dispersion, especially for those significant positive points of PD, FRAC_AM, and SPLIT and negative points of SHDI located in the urban areas of Figure 6. In the future, we can also gradually improve the green space within the metropolitan area and its connectivity with the urban–suburban area, reduce the dispersion of the green space landscapes, and ensure the biological migration and material energy cycle to alleviate the ecological pressure from the metropolitan area. (III) In rural areas, we can strengthen the cultural promotion, management, protection, and restoration surrounding large parks and wetlands to avoid unnecessary damage from socioeconomic activities. Because it helps improve the connectivity between high-quality landscapes, reduce the PD, FRAC_AM, and SPLIT, and improve landscape diversity (SHDI), thus promoting the stability and improvement of ecosystem services (Figure 6).

## 5. Conclusions

This study proposed a novel framework based on the InVEST and STWR models to explore the spatiotemporal heterogeneities between suburban landscape patterns and ecosystem services in urban–suburban areas of Fuzhou, China. Several findings can be drawn from our research. Firstly, ecosystem services' response to landscape patterns has spatial and temporal heterogeneity rather than just urban–rural differences. Analyzing the heterogeneities based on our proposed framework helps provide new insights into enhancing ecosystem services by shaping landscape patterns. Second, urban–suburban areas, especially those close to the boundaries, are crucial for ecosystem services' response to changes in landscape patterns. Notably, dynamic changes in the urban–suburban boundary are critical to the relationships between LPIs and CES. Finally, although the urban–suburban areas are critical, we found that the coefficients of LPIs on CES were smaller than those of urban areas. The coefficient weakening may be due to more high-quality suburban

landscapes "resisting" urbanization. This empirical study is just one case to verify the effectiveness of our framework. We hope the proposed framework can be applied in many related disciplines.

**Author Contributions:** X.Z. (Xinyan Zou): data curation, visualization, writing—original draft. C.W.: writing—review and editing, validation. X.Q.: conceptualization, methodology, funding acquisition, supervision, writing—original draft, writing—review and editing. X.M.: funding acquisition, writing—review and editing. Z.W.: writing—review and editing. Q.F.: data curation, writing—review and editing. Y.L.: writing—review and editing. X.Z. (Xinhan Zhuang): data curation, writing—review and editing. All authors have read and agreed to the published version of the manuscript.

**Funding:** This work is financially supported by the National Natural Science Foundation of China (42202333), the U.S. National Science Foundation (Grant No. 2019609), the Natural Science Foundation of Fujian Province (2021J05030 and 2022J01152), and the Science and Technology Innovation Project of Fujian Agriculture and Forestry University (KFB23150 and KCX21F33A).

**Institutional Review Board Statement:** Not applicable.

**Informed Consent Statement:** Not applicable.

**Data Availability Statement:** Data will be made available on request.

**Conflicts of Interest:** The authors declare no conflicts of interest.

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
