# Peer review of "Spatiotemporal Heterogeneous Responses of Ecosystem Services to Landscape Patterns in Urban–Suburban Areas"

_sustainability, doi:10.3390/su16083260_

Round 1

Reviewer 1 Report

Comments and Suggestions for Authors

Dear authors,

I am sending several ideas and comments regarding Your article:

1.      The research is very relevant because the development of social and economic infrastructure often has a negative impact on environmental public goods and ecosystems. The quality of ecosystem services can significantly decrease due to urbanization. This leads to the loss of natural resources and landscapes, and at the same time to the increasingly important political debates about their conservation. However, the study lacks an analysis of specific policy measures that could address these issues. How to stop the degradation of ecosystem services in the urban and sub-urban areas of Fuzhou remains an open question, requiring urgent solutions. It is also unclear what interest groups should be involved in solving the problem.

2.      Specific insights and proposals on how to improve ecosystem services through landscape modeling are also lacking.

3.      The purpose of the study should be more precisely defined in the abstract.

Author Response

Please see our attached response document. Thank you for your valuable comments.

Reviewer 2 Report

Comments and Suggestions for Authors

The paper discussing the Spatiotemporal heterogeneous responses of ecosystem services to landscape patterns in urban-suburban areas seems technically sound. However, a few potential areas might require clarification and improvement. 

In the abstract section mention:

What is the proposed framework for exploring spatiotemporal heterogeneity in urban-suburban areas?

What is the potential utility of this framework for decision-making in improving ecosystem services around cities?

In the Introduction section mention 

How has the city of São Paulo, Brazil, been affected by environmental problems due to urbanization?

What findings were reported regarding the relationship between landscape pattern indices and ecosystem services in Baguio City and Jakarta, Indonesia?

3.1. CES Assessment section mention

What trend was observed in the spatial distribution of CES from 2000 to 2020?

In Discussion mention

According to the empirical studies, how did the impact of Landscape Pattern Indices (LPIs) on Comprehensive Ecosystem Service (CES) change over time in urban-suburban areas?

What is the significance of the negative impact of the SPLIT index on CES, and how does it relate to urban expansion?

What implications arise from the significant points being close to the boundaries of urban-suburban areas in terms of CES response to LPIs during urbanization?

Author Response

(The authors gave the same response as above.)

Reviewer 3 Report

Comments and Suggestions for Authors

Dear authors,

I appreciate the study is up-to-date and presents mathematical models used in the evaluation of ecosystems.

I want to ask the authors if such studies have practical applicability in China with concrete results that can improve the eventual degradation of the ecosystems that are the subject of the study?

I would suggest the authors in the introduction to mention, if it is the case in China and the study area, about the invasive species of plants and insects that degrade urban ecosystems.

I would suggest a better highlighting of the objectives of the study and whether the proposed model can be implemented in other regions of China.

It is not clear from the research method which are the ecosystems studied, what kind of forests are mentioned, which are the species of trees, shrubs, or plants from the spontaneous flora.

Water sources are also mentioned, the rivers in the region were studied, I would suggest that they be mentioned, which are the analysed parameters.

The study of the carbon footprint is mentioned, but it is not clear what were the methods used in the research and the results obtained.

Author Response

(The authors gave the same response as above.)
